# The Knockdown of TREK-1 in Hippocampal Neurons Attenuate Lipopolysaccharide-Induced Depressive-Like Behavior in Mice

**DOI:** 10.3390/ijms20235902

**Published:** 2019-11-24

**Authors:** Ajung Kim, Hyun-Gug Jung, Yeong-Eun Kim, Seung-Chan Kim, Jae-Yong Park, Seok-Geun Lee, Eun Mi Hwang

**Affiliations:** 1Center for Functional Connectomics, Korea Institute of Science and Technology (KIST), Seoul 02792, Korea; kitkim819@kist.re.kr (A.K.); wjdjd77@kist.re.kr (H.-G.J.); kye93@kist.re.kr (Y.-E.K.); 216008@kist.re.kr (S.-C.K.); 2KHU-KIST Department of Converging Science and Technology, Graduate School, Kyung Hee University, Seoul 02447, Korea; 3School of Biosystem and Biomedical Science, College of Health Science, Korea University, Seoul 02841, Korea; jaeyong68@korea.ac.kr; 4Division of Bio-Medical Science and Technology, KIST School, University of Science and Technology (UST), Seoul 02792, Korea; 5Department of Science in Korean Medicine, Kyung Hee University, Seoul 02447, Korea

**Keywords:** TREK-1, depression, hippocampus, neurotrophic factor, conditional knockdown

## Abstract

TWIK-related potassium channel-1 (TREK-1) is broadly expressed in the brain and involved in diverse brain diseases, such as seizures, ischemia, and depression. However, the cell type-specific roles of TREK-1 in the brain are largely unknown. Here, we generated a Cre-dependent TREK-1 knockdown (Cd-TREK-1 KD) transgenic mouse containing a gene cassette for Cre-dependent TREK-1 short hairpin ribonucleic acid to regulate the cell type-specific TREK-1 expression. We confirmed the knockdown of TREK-1 by injecting adeno-associated virus (AAV) expressing Cre into the hippocampus of the mice. To study the role of hippocampal neuronal TREK-1 in a lipopolysaccharide (LPS)-induced depression model, we injected AAV-hSyn-BFP (nCTL group) or AAV-hSyn-BFP-Cre (nCre group) virus into the hippocampus of Cd-TREK-1 KD mice. Interestingly, the immobility in the tail suspension test after LPS treatment did not change in the nCre group. Additionally, some neurotrophic factors (BDNF, VEGF, and IGF-1) significantly increased more in the nCre group compared to the nCTL group after LPS treatment, but there was no difference in the expression of their receptors. Therefore, our data suggest that TREK-1 in the hippocampal neurons has antidepressant effects, and that Cd-TREK-1 KD mice are a valuable tool to reveal the cell type-specific roles of TREK-1 in the brain.

## 1. Introduction

TWIK-related potassium channel-1 (TREK-1) is one of the two-pore-domain potassium (K2P) channels involved in the background leakage of potassium ions, and thus, influences resting membrane potential and cell excitability [1]. TREK-1 is involved in various neuropathological conditions, such as seizures, ischemia, and depression [2,3]. Particularly, there are several previous studies that investigated the association between TREK-1 and depression, and inhibitors of TREK-1 have been considered to have antidepressant effects. Clearly, antidepressant effects have been reported in both TREK-1-deficient mice and mice treated with fluoxetine, curcumin, spadin, SID1900, and TKDC, which are well-known TREK-1 inhibitors [2,4,5,6,7,8].

To date, the antidepressant effects of TREK-1 have been explained by an increase in the amount of serotonin in the brain through the increased neuronal activity of serotonin neurons [2,7]. However, TREK-1 is known to be well expressed in serotonin neurons and neurons of the prefrontal cortex (PFC) and hippocampus, which are important brain regions associated with the cognitive aspects of depression [9,10,11]. Additionally, the serotonin hypothesis is still controversial in the studies involving humans with major depressive disorder [12], and several magnetic resonance imaging-based studies have reported reduced neuron volumes in the PFC and hippocampus [13,14]. Thus, to better understand the antidepressant effects of TREK-1 channels, identifying the effects of TREK-1 on other neurons, specifically the PFC or hippocampus, is significantly useful.

Depression has a variety of factors, one of which is depression caused by chronic inflammatory components [15,16,17]. Symptoms of depression are often caused by pro-inflammatory cytokines, mainly due to tumor necrosis factor α (TNF-α), interleukin 1 beta (IL-1β), and interleukin 6 (IL-6) [18,19,20]. A commonly used model to reproduce this inflammatory-mediated depression is the treatment of lipopolysaccharides (LPSs), known as cytokine inducers. Animal models treated with LPS showed increased depression-related behavior and exhibit increased immobility, particularly in forced swim test (FST) and tail suspension test (TST) [21,22]. Additionally, before such depression-related behavior increases, sickness-related behavior increases and locomotor activity decreases [23]. In general, sickness behavior is reported to be at maximum between 2 and 6 h after LPS injection; subsequently, sickness behavior decreases, and depressive symptoms are observed for approximately 24 h [24,25].

The neurotrophic factor is associated with depressive symptoms. Particularly, there is increasing evidence that decreased expression of brain-derived neurotrophic factor (BDNF) and vascular endothelial growth factor (VEGF) in the brain is associated with neuropsychiatric disorders [26]. Moreover, LPS-induced pro-inflammatory cytokines reduce the expression of BDNF and VEGF [26,27]. BDNF and its receptor, Ntrk2, increase the expression in the hippocampus when treated with antidepressants, and increased VEGF was observed during exercise, which has an antidepressant effect [28,29,30]. The centrally administered insulin growth factor-1 (IGF-1) has no effect on LPS-induced sickness behavior, although it results in decreased depressive-like behavior [31]. Increasing reports of various neurotrophic factors associated with depression are considered to be an important cellular-level molecular mechanism to determine the therapeutic effects of depression.

In the present study, we reported for the first time a Cre-dependent TREK-1 knockdown (Cd-TREK-1 KD) transgenic mouse in which the TREK-1 short hairpin ribonucleic acid (shRNA) was expressed by a Cre-dependent manner. This mouse model can be used to study the function of cell type- or region-specific TREK-1 using various Cre viruses or Cre mouse lines. Indeed, by using mice that selectively inhibited TREK-1 in hippocampal neurons, we found that inhibiting only TREK-1 present in hippocampal neurons could provide antidepressant effects. Additionally, the antidepressant effect caused by TREK-1 knockdown in hippocampal neurons could be mediated by preventing the reduction of neurotrophic factors (BDNF, VEGF, and IGF-1) driven by LPS.

## 2. Results

### 2.1. The Hippocampal Infection of Cre Virus Inhibits the Expression of TWIK-Related Potassium Channel-1 (TREK-1) Selectively

According to our previous study, we generated a pSico-Red system containing TREK-1 shRNAs, which has a Cre-dependent gene-knocking down vector system [32]. This pSico-Red system was able to identify cells that express shRNA, which was not possible with the original pSico systems. This system can also provide a gene cassette capable of easily producing a Cre-dependent gene knockdown transgenic mouse. Thus, we generated transgenic mice using pSico-Red vectors with validated TREK-1 shRNA sequences for cell type- or region-specific functional studies of TREK-1. As shown in Figure 1A, we added an elongation factor 1 alpha (EF1α) promoter and mCherry that was not originally present in the pSico system for a separate mCherry expression. The genotype was further confirmed by genomic deoxyribonucleic acid polymerase chain reaction (DNA PCR) with specific primer sets designed with EF1α promoter and mCherry site. Gel electrophoresis of the PCR products showed one band at 516 bp (Figure 1B), indicating pSico-Red-shTREK-1. Fluorescent proteins (green fluorescent protein [GFP] and mCherry) and TREK-1 expressions were confirmed by western blot analysis (Figure 1C). Moreover, we further confirmed pSico-Red-shTREK-1 mice generally expressed GFP and mCherry protein in several brain regions, such as the cerebral cortex, cerebellum, medial PFC (mPFC), hippocampus, and striatum (Appendix A).

To investigate the Cre-inducible TREK-1 shRNA-expressing cells in our transgenic mice line, we injected adeno-associated virus (AAV) expressing mCherry only (AAV-EF1α-mCherry, Control [CTL] virus) or mCherry-IRES-Cre (AAV-EF1α-mCherry-IRES-Cre, Cre virus) into the dentate gyrus (DG) of pSico-Red-shTREK-1 mice (Figure 2A) because a previous report confirmed that TREK-1 is highly expressed in the DG of the hippocampus [10]. We expected this mouse to show GFP and mCherry fluorescence and strong TREK-1 expression in cells infecting CTL virus and still mCherry fluorescence and reduced TREK-1 expression in cells infecting Cre virus (Figure 2B). Three weeks after AAV injection, proper AAV injections and reduced TREK-1 expression in the DG were confirmed by fluorescent imaging (Figure 2B). We used a commercially available EF1α promoter-driven mCherry-IRES-Cre virus for the validation of pSico-Red-shTREK-1 mice and low light power to distinguish between high-expression mCherry of the virus (high copy) and endogenous low-expression mCherry of this mouse (low copy) (Appendix A). The levels of TREK-1 messenger RNA (mRNA) and protein were significantly reduced by Cre virus in pSico-Red-shTREK-1 mice. In contrast, there was no difference in the expression of mRNA and protein of TREK-1 in the non-injected region (Figure 2C–G). These results show that pSico-Red-shTREK-1 mice significantly reduce Cre-specific TREK-1 expression in the hippocampus.

### 2.2. TREK-1 Is Upregulated by Lipopolysaccharide (LPS) in the Hippocampus

We successfully applied these pSico-Red-shTREK-1 mice to be knocked down on a Cre-dependent manner. Subsequently, using this TREK-1 conditional knockdown system, we investigated whether neuronal TREK-1 expression was associated with acute depression. Because there was a report that the expression of TREK-1 in the PFC increased in rats under chronic mild stress conditions, but not in the hippocampus [33], we need to confirm TREK-1 expression in the hippocampus of the LPS-induced acute depression model. LPS-induced depression model is one of the frequently used animal models for the study of depression [34,35]. To investigate the effect of TREK-1 in acute depression-like behavior s induced by LPS in mice, the mice were injected with AAV-hSyn-BFP (neuronal CTL, nCTL) or AAV-hSyn-BFP-Cre (neuronal Cre, nCre) into the DG of the hippocampus (Figure 3A,B). We used hSyn promoter to knock down TREK-1 specifically in the neurons of the DG. After three weeks, LPS (1.2 mg/kg) or a saline was administered. As shown in Figure 3C, most of the Cre-injected cells in the DG only expressed mCherry signal except GFP signal. Moreover, it was confirmed that TREK-1 expression was significantly reduced in the cells infected with hSyn-BFP-Cre-virus. The LPS-treated group significantly induced TREK-1 expression levels (Figure 3C). mRNA and protein levels of the TREK-1 were affected by LPS (Figure 3D,E). Considering these data, we confirmed that mRNA and protein levels of TREK-1 were upregulated by LPS in the hippocampus.

### 2.3. Neuronal TREK-1 Knockdown in the Dentate Gyrus Reduced Depression-Like Behaviors Induced by LPS in Mice

Subsequently, we measured bodyweight changes at 4 and 24 h after LPS injection to verify sickness behavior s and depression-like behavior s observed in the LPS-induced depression model. These sickness behaviors are revealed to occur at the stage of pro-inflammation, reaching a maximum of 2–6 h after the injection of LPS and decreasing thereafter [23,24,25,36]. The bodyweight of the mice was significantly reduced after LPS administration regardless of time point. Moreover, there was no difference between the nCTL and nCre groups (Figure 4A). An open field test (OFT) was subsequently performed 4 and 24 h after LPS injection, respectively (Figure 4B). After 4-h LPS treatment, the movement speed of the mice showed a significant decrease in all LPS treatment groups regardless of Cre (Bonferroni’s post hoc; *p* < 0.0001) (Figure 4C). The total distance moved by the mice also showed a similar trend (Figure 4D). However, the decreased locomotor activity was recovered 24 h after LPS treatment in all groups. These results showed that our LPS-induced mouse model reproduces the sickness behavior -induced characteristics of LPS as previously known and confirmed that the reduction of neuronal TREK-1 does not change this behavior.

We subsequently performed a TST 24 h after LPS treatment for the assessment of LPS-induced depressive-like behavior. Interestingly, immobility was significantly reduced in the LPS-treated nCTL in the TST test (Bonferroni’s post hoc; *p* = 0.0003), but there was no significant difference in the LPS-treated nCre mice (Bonferroni’s post hoc; *p* = 0.7164) (Figure 4G). The increased immobility by LPS was recovered by Cre-dependent TREK-1 reduction despite the presence of LPS (Bonferroni’s post hoc; *p* = 0.0002). In the LPS-treated-nCre (TREK-1 cKD) group, struggling time was significantly increased relative to the mice of the LPS-treated nCTL group (Bonferroni’s post hoc; *p* = 0.0001) (Figure 4H). These results indicate that the knockdown of neuronal TREK-1 in the DG of the hippocampus significantly reduced the depressive-like behavior al effects of LPS. These anti-depressive-like phenotypes were also reproduced in restraint stress model, which is one of the general depression mice models (Appendix A). Taken together, we confirmed that our hippocampal DG-specific neuronal TREK-1 cKD mice had antidepressant effects in various depression models.

### 2.4. Neuronal TREK-1 Did Not Influence LPS-Induced Cytokine Expression

Since the previous studies have shown that TREK-1 deficiency reduces cytokine secretion in endothelial cells [37], we investigated whether the reduction of neuronal TREK-1 itself affected the inflammatory response to LPS. Moreover, several studies have described the association between inflammation and depression [38,39]. Thus, we performed cytokine profile array experiments to identify various cytokines in the hippocampal tissues simultaneously. As shown in Figure 5A, LPS induced a significant upregulation of the expression of cytokine and chemokines, including IL-1β, IL-6, and TNF-α in all groups. However, the decrease of TREK-1 by Cre had no effect. These results were confirmed again in the real-time PCR experiment. mRNA levels of IL-1β and TNF-α were insignificantly different between the nCTL and nCre groups in Cd-TREK-1 KD mice (Figure 5B,C). Thus, these results suggest that hippocampal DG-specific knockdown of neuronal TREK-1 did not alter the inflammatory response by LPS.

### 2.5. Neuronal Inhibition of TREK-1 in the Hippocampus Significantly Prevented the Decrease of Neurotropic Factors in LPS-Induced Depression Mice

To determine the mechanism of the antidepressant effects of neuronal TREK-1 inhibition in the hippocampus, we investigated the levels of corticosterone and various neurotrophic factors. Because most antidepressants have been shown to lower cortisol levels and promote neurotrophic factors, such as BDNF in both major depressive disorder patients and animal models [26,40], we first identified the amount of serum corticosterone known to increase with stress in nCTL and nCre mice. Serum corticosterone levels were similar before LPS treatment, but rapidly increased after LPS treatment, and the effect was suppressed in the nCre group (*p* < 0.0001) (Figure 6A). Next, we investigated the expression of various neurotrophic factors using virus-infected hippocampal tissues. VEGF levels in the hippocampal tissues were significantly reduced in the LPS-treated group, but this effect was restored again in the nCre group (*p* = 0.0454) (Figure 6B). The expression of IGF-1 and BDNF mRNAs, which promote neural stem cell proliferation, was similar to VEGF protein expression patterns in each group (Figure 6C,D). Protein levels of IGF-1 and BDNF had similar patterns in the LPS-treated group (IGF-1; *p* = 0.0011, BDNF; *p* = 0.0287) (Figure 6E–G). To confirm the expression changes of neurotrophic factors by LPS in brain regions not infected with the virus, expression was also examined in the mPFCs, which also have high levels of TREK-1 in the brain. As expected, the hippocampal nCre virus had no effect on the mPFC region (Appendix A). Since the expression of the receptors may also have the same biological effects as increased ligand expression, we have examined the expression of neurotrophic factor receptors. The mRNA and protein expression of VEGF receptor-1 (VEGFR-1) and VEGF receptor-2 (VEGFR-2) in the hippocampus was not significantly different in all groups (Figure 6H,I,L,M). The mRNA and protein expression of IGF-1r was significantly reduced by LPS in the nCTL groups, but there is no significant difference in the nCre groups (Figure 6J,N). The mRNA level of Ntrk2 was significantly reduced by LPS, but its protein level had no significant difference in all groups (Figure 6K,O). Taken together, we suggest that the antidepressant effect of the hippocampal neuronal TREK-1 reduction was obtained by inhibiting the reduction of neurotrophic factors, such as BDNF, IGF-1, and VEGF.

## 3. Discussion

This is the first report to study the role of TREK-1 in acute depression by regulating the expression of TREK-1 in a cell type-specific manner. We created a transgenic mouse in a short time using the pSico-Red-shTREK-1 sequence, which easily distinguishes the TREK-1 shRNA-expressing cells by fluorescence, developed in the previous study [32]. We named these mice as Cd-TREK-1 KD mice because they expressed TREK-1 shRNA in a Cre-dependent manner and effectively reduced TREK-1. Additionally, pSico-Red-shTREK-1 expression is comparable in many areas of the brain, including the hippocampus; hence, the mouse can be applied to future TREK-1 functional studies in other areas. Using this transgenic mouse and neuron-specific Cre virus, neuronal TREK-1 was selectively reduced in the hippocampal DG region, and we identified antidepressant behavior s similar to overall TREK-1 reduction.

Suppression of overall brain TREK-1 in TREK-1 knockout mice or treatment with TREK-1 inhibitors has shown effective antidepressant effects [2,4,5,6,7,8]. Indeed, TREK-1 is considered a background potassium channel, and inhibition of this channel can increase the activity of the expressing neurons. Therefore, several scientists have believed that serotonin secretion, an important target for depression, is increased by suppressing TREK-1 in serotonin neurons. However, the expression of TREK-1 in serotonin neurons in the dorsal raphe is lower than that of other K2P channels, TASK-3 [10], and there has been no report on the antidepressant effects of TREK-1 inhibition in these neurons alone. The association between serotonin levels and antidepressant effects is also controversial [12].

The DG of the hippocampus is an important region of the brain that determines the effectiveness of treatment for depression. Certainly, antidepressants have been shown to increase neural stem cells and adult neurogenesis in the DG [2,41]. Chronic antidepressant treatment increases BDNF expression in the hippocampus; in particular, DG-specific BDNF infusion also has an antidepressant effect [30,42,43]. In this study, we directly infected neuron-specific Cre virus in the DG to rule out the effect of serotonin neurons projected from the dorsal raphe. Interestingly, antidepressant behavior was observed in both acute and chronic depression models with DG neuron-specific TREK-1 inhibition, indicating that the antidepressant effect is sufficient to enhance DG neuron activity alone. More specifically, the suppression of excitability of neonatal neurons in DG by the use of chemogenetic approaches eliminates the antidepressant effects of fluoxetine without altering adult neurogenesis [44]. Certainly, because the DG has a variety of neuronal cell types and their activity is regulated between each other, a more detailed study on neuronal types associated with TREK-1 in depression is required.

The LPS-induced depression model is a well-known acute depression model, in which microglia-secreting cytokines activated by LPS cause depressive behavior by decreasing BDNF expression [41,42]. Our LPS-induced depression model also showed sick behavior at 4 h and recovered at 24 h, as is known. Additionally, these mice reproduced well, increasing corticosterone levels and TST immobilization at 24-h LPS treatment. However, we found a possible increase in immobility in another depression behavior al test, FST, but there was no statistically significant difference. TST is more suitable for depression behavior than FST because it maintains immobility longer than that of FST [45]. Additionally, since our cKD mice induced TREK-1 reduction only in neurons without affecting the microglia or astrocytes, as expected, there was no effect on cytokine secretion by LPS.

Next, we examined the changes in the expression of neurotrophic factors known to be associated with depressive behavior. Interestingly, BDNF, IGF-1, and VEGF, all of which have antidepressant effects, increased in nCre mice than in nCTL mice under LPS treatment. These results indicate that the inhibition of TREK-1 increases neurotrophic factors expression, thereby increasing neuronal plasticity in the short term and increasing adult neurogenesis in the long term. Additionally, inhibition of TREK-1 in neuronal cells induces depolarization-induced Ca^2+^ influx and activates PI3K/AKT and extracellular signal-regulated kinase (ERK) signaling [46]. Thus, reduction of TREK-1 in the nCre groups of our mice may also increase the basal activity of the AKT and ERK signaling, which may be effective in preventing BDNF reduction in inflammatory responses. Finally, since the expression of neurotrophic factor receptors may also alter the effects of neurotrophic factors, the expression of each receptor was examined, but no significant changes by LPS were observed.

Therefore, our results suggest that hippocampal neuronal TREK-1 serves sufficiently as a target for antidepressants. Furthermore, hippocampal DG neurons can significantly increase neurotrophic factors, such as BDNF, IGF-1, and VEGF in inflammatory environments by TREK-1 inhibition. These results suggest that TREK-1 in the hippocampal neurons has antidepressant effects, and the Cd-TREK-1 KD mice will be a valuable tool to reveal the cell type-specific roles of TREK-1 in the brain.

## 4. Materials and Methods

### 4.1. Plasmid Preparation

The TREK-1 nucleotides from 998 to 1018 (5′–gcgtggagatctacgacaagt–3′) was selected for target region of TREK-1 shRNA as previously described [32,47]. To apply the pSico-Red system, we synthesized the following sequence and inserted it into the pAAV-Sico-Red vector: 5′–tgcgtggagatctacgacaagtttcaagagaacttgtcgtagatctccacgcttttttc–3′ and 5′–tcgagaaaaaagcgtggagatctacgacaagttctcttgaaacttgtcgtagatctccacgca–3′. The annealed double-stranded oligo was inserted into HpaI-XhoI restriction enzyme site of pAAV-Sico-Red and verified by sequencing.

### 4.2. Generation and Genotyping of pSico-Red-shTREK-1 Transgenic Mice

To obtain TREK-1 conditional knockdown mice, we first generated transgenic mice containing from the EF1α promoter to the TREK-1 shRNA in pAAV-Sico-Red-TREK-1 shRNA vector (pSico-Red-shTREK-1 mice). In more detail, pAAV-Sico-Red-TREK-1 shRNA vector was linearized with MluI and PmlI to isolate the transgenic cassette. The isolated cassette was inserted into C57BL/6 mouse embryos (fertilized one-cell zygotes) and then implanted into female mice (Marcrogen Inc., Seoul, Korea). pSico-Red-shTREK-1 mice were genotyped with genomic DNA PCR analysis using genomic DNA prepared from their tails, and wild type mice were used as a control. pSico-Red-shTREK-1 forward (EF1α pro): 5′–ccacacaaaggaaaagggcc–3′, pSico-Red-shTREK-1 reverse (mCherry-N): 5′–ggtggccccctgcccttcgcc–3′. Animal care and handling were performed according to the institutional guidelines of Institutional Animal Care and Use Committee (IACUC-2017-056, 6 May 2017) at the Korea Institute of Science and Technology (Seoul, Korea).

### 4.3. Stereotaxic Injection

Mice (6–7 weeks old) were anaesthetized with an Avertin^®^ (2,2,2-tribromethanol in 2-methyl 2-butanol) and placed in a stereotaxic frame. Briefly, the scalp was opened, and two holes were drilled in the skull (−1.8 mm AP from bregma, ± 1.6 mm ML). AAV-EF1α-mCherry and AAV-EF1α-mCherry-IRES-Cre (AAV5) were purchased from UNC Vector Core. AAV-hSyn-BFP and AAV-hSyn-BFP-Cre were packaged with serotype DJ at KIST Virus Facility. These viruses were bilaterally injected (250 nL per side) into the dentate gyrus area (2.1 mm DV from the dura) through a Hamilton Syringe with a syringe pump (KD Scientific, Holliston, MA, USA) that infused the virus at a speed of 0.1 µL/min. At the injected points, Hamilton Syringe was left in place for 10 min. After injection, the syringe stayed in target place for an additional 10 min.

### 4.4. LPS Treatment

LPS was purchased from Sigma-Aldrich (St. Louis, MO, USA). Mice were injected intraperitoneally with either LPS (1.2 mg/kg) or vehicle (sterile saline) [22]. After injections, mice were measured a body-weight.

### 4.5. Behavior Test (OFT, TST)

Behavioral tests were performed after the two weeks of virus injection. Tests were conducted in a soundproofing room with dim light of 10 lux. Every test was analyzed with the Ethovision 3.0 software (Noldus, San Diego, CA, USA).

#### 4.5.1. Open Field Test (OFT)

To evaluate the effects of LPS on exploratory locomotor activity, mice were tested 4, 24 h after treatment. The OFT was performed according to previous methods [21], using an OFT box (30 × 30 × 39 cm). Mice were placed in the center of the OFT box and activity was recorded for a period of 12 min and evaluated at last 10 min.

#### 4.5.2. Tail Suspension Test (TST)

Mice were suspended by the end of their tail with tape (about 50 cm above the floor) and suspended above the floor of a white acrylic box. We recorded the immobility time during a 6 min.

### 4.6. Western Blots

For western blotting, hippocampus and medial prefrontal cortex tissues of mice were lysed with lysis buffer (50 mM HEPES, 0.1% Sodium deoxycholate, 1% Triton X-100, 1 mM PMSF and 0.1% SDS containing protease inhibitor cocktail, pH 7.4). Total protein was subjected to SDS-PAGE and transferred to PVDF membranes. The membranes were blocked using 5% non-fat milk, and then, blotted with the appropriate antibodies: Anti-GFP (Santa Cruz, Dallas, TX, USA, sc-9996), anti-mCherry (abcam, ab167453, Cambridge, UK), anti-TREK-1 (Santa Cruz, sc-11556), anti-BDNF (Thermo Fisher, #710306), anti-IGF-1 (abcam, ab9572), anti-IGF-1r (Cell signaling, #9750, Danvers, MA, USA), anti-VEGFR2 (Cell signaling, #2479) and anti-Actin (Sigma-Aldrich, A5441). The membranes were then washed and incubated with HRP-conjugated secondary antibodies (Jackson ImmunoResearch, West Grove, PA, USA).

### 4.7. Immunohistochemistry

Mouse brain were obtained following perfusion with 4% PFA in PBS solution, and 40 µm-thick free-floating sections were prepared using a vibratome. Sections were permeabilized with 0.5% Triton X-100 in PBS for 20 min at RT, followed by blocking with 5% donkey serum and 2% BSA with 0.3% Triton X-100 in PBS for 1 h at RT. Tissues were incubated overnight at 4 °C with primary antibodies. The following antibodies were used in this study: Anti-TREK-1 (Alomone labs, APC-047, Jerusalem, Israel). For detection, suitable Alexa fluor-tagged secondary antibody (Jackson ImmunoResearch) was used, and the tissues were counterstained with DAPI.

### 4.8. VEGF ELISA

Mice hippocampus and medial prefrontal cortex samples were homogenized with lysis buffer (50 mM HEPES, 0.1% Sodium deoxycholate, 1% Triton X-100, 1 mM PMSF and 0.1% SDS containing protease inhibitor cocktail, pH 7.4). Levels of VEGF was measured in the tissue lysate samples using a VEGF ELISA kit (R&D Systems, MMV00, Minneapolis, MN, USA). Absorbance was measured using Infinite M200 Pro (TECAN, Männedorf, Switzerland), and the concentrations of VEGF was determined based on a standard curve according to the manufacturer‘s instruction.

### 4.9. Serum Corticosterone ELISA

The serum corticosterone levels using an ELISA kit (Enzo Life Sciences, Inc, Farmingdale, NY, USA) according to the manufacturer‘s instruction. Serum was obtained from trunk blood. After blood collection, samples could keep at room temperature for 30 min. The blood sample was then centrifuged at 4000 rpm for 10 min, and the serum was transferred to a micro tube and stored at −80 °C until corticosterone ELISA assay was conducted.

### 4.10. RT-PCR and Real-Time PCR

Total RNA was isolated from several mouse tissues (hippocampus and prefrontal cortex) using an RNA purification Kit (GeneAll) according to the manufacturer’s instruction. The cDNAs were synthesized from 1 mg total RNA and reverse transcription was performed using a SensiFASTTM cDNA Synthesis Kit (BIOLINE) according to the manufacturer’s instruction. RT-PCR primer sequences were as follows: TREK-1: Forward, 5′–GTCCTCTACCTGATCATCGGAGC–3′; reverse, 5′–CCTAGCTGATCACCAACCCC–3′, GAPDH: Forward, 5′–GTCTTCACCACCATGGAGAA–3′; reverse, 5′–GCATGGACTGTGGTCATGAG–3′. GAPDH was used as a loading control. Real-time PCR was performed using a SensiFAST^TM^ Probe Hi-ROX kit (BIOLINE, London, UK). Primer sets for TNF-α (Mm.PT.58.12575861), IL-1β (Mm.PT.58.41616450), IGF-1 (Mm.PT.58.5811533), BDNF (Mm.PT.58.8157970), TREK-1 (Mm.PT.58.13878544), VEGFR-1 (Mm.PT.58.43852013), VEGFR-2 (Mm.PT.58.5869721), IGF1r (Mm.PT.58.11619137), Ntrk2 (Mm.PT.58.11070732) and GAPDH (Mm.PT.39a.1) were purchased at IDT (PrimeTime qPCR assays). GAPDH was used as a reference gene. The 2^−ΔΔ*Ct*^ method was used to calculate fold changes in gene expression. All experiments were repeated at least three times at triplicates.

### 4.11. Cytokine Profile Array

Levels of cytokines in the mouse hippocampus tissues were measured using cytokine profile (R&D Systems, ARY006). Tissue was lysed and processed as per the manufacturer‘s instructions. Results were analyzed using ImageJ software (version;1.8.0, Bethesda, MD, USA).

### 4.12. Statistics

All data are expressed as the mean ± standard error of the mean (S.E.M.). Data were analyzed by a two-way ANOVA followed by Bonferroni‘s post hoc test and Turkey‘s post hoc test using the GraphPad Prism 8.0 software. Differences were considered statistically significant when *p* < 0.05.

## Figures and Tables

**Figure 1 ijms-20-05902-f001:**
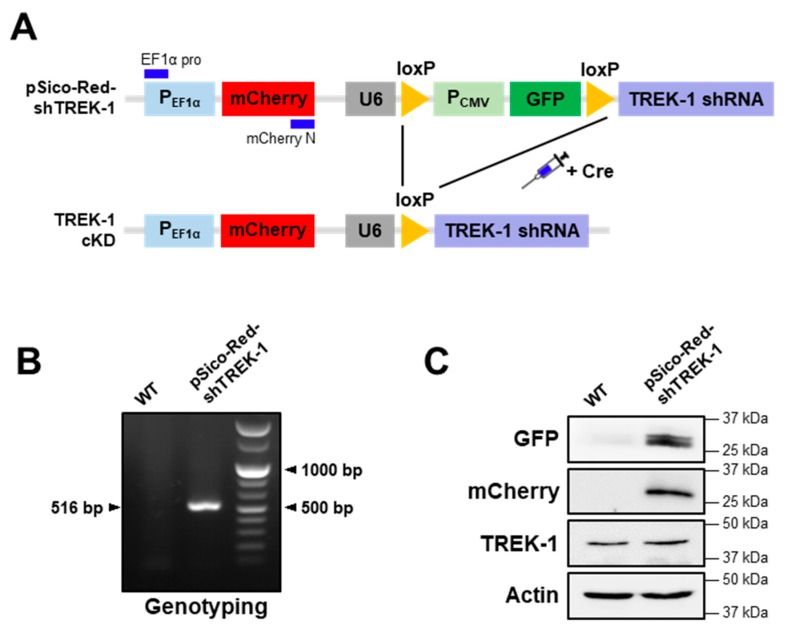
Generation of transgenic mice for the Cre-dependent knockdown of TWIK-related potassium channel-1 (TREK-1). (**A**) Scheme of the mice model of TREK-1 conditional knockdown. (**B**) Genotypes of mice were identified using genomic deoxyribonucleic acid. Primers elongation factor 1 alpha pro and mCherry-N were used to distinguish the targeted allele (515 bp) from the untargeted wild-type allele. (**C**) Protein samples from mice hippocampus were tested for the protein expression of green fluorescent protein, mCherry, and TREK-1 by western blot analysis. Actin was used as a loading control.

**Figure 2 ijms-20-05902-f002:**
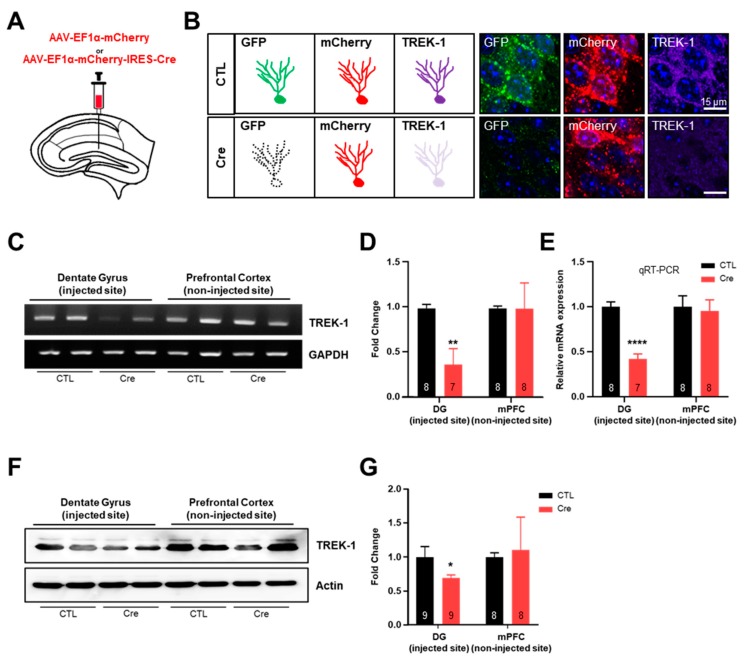
Validation of the hippocampal TWIK-related potassium channel-1 (TREK-1) conditional knockdown (cKD) mice. (**A**) An illustration of a hippocampal slice of TREK-1 cKD mice to show the site of adeno-associated virus (AAV)-elongation factor 1 alpha (EF1α)-mCherry (Control [CTL]) or AAV-EF1α-mCherry-IRES-Cre (Cre) injection. (**B**) Scheme representation of green fluorescent protein, mCherry, and TREK-1 under CTL or Cre virus infection. TREK-1 signals were compared by immunohistochemical staining in the dentate gyrus injected with CTL or Cre virus in pSico-Red-shTREK-1 mice. Validation of TREK-1 cKD efficiency of messenger ribonucleic acid levels by real-time polymerase chain reaction (RT-PCR) (**C**–**E**) and quantitative RT-PCR ©. Validation of TREK-1 cKD efficiency of protein levels by Western blotting (**F**,**G**). The numbers inside each bar indicate the number of samples. Data are presented as means ± standard error of the mean (* *p* < 0.05, ** *p* < 0.01, **** *p* < 0.0001).

**Figure 3 ijms-20-05902-f003:**
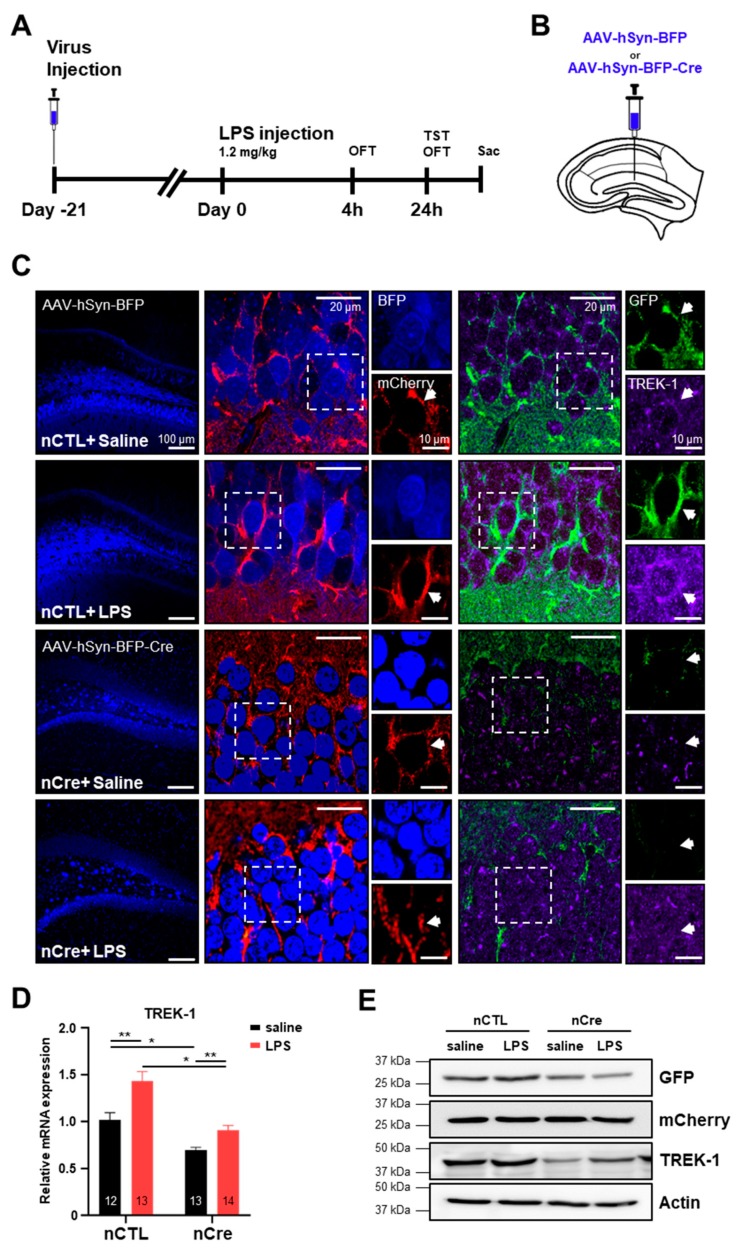
Lipopolysaccharide (LPS) increases the expression of TREK-1 in the hippocampus. (**A**) Experimental procedure for the LPS injection test schedule. Viruses were injected into the bilateral dentate gyrus, followed by a 21-day recovery (Day −21). LPS (1.2 mg/kg) or its vehicle was administered 1 time (Day 0), and subsequently, 4 h later the open field test (OFT) and 24 h later the OFT and tail suspension test were performed. (**B**) An illustration of a hippocampal slice of pSico-Red-shTREK-1 mice showing the site of AAV-hSyn-BFP (neuronal control) or AAV-hSyn-BFP-Cre (neuronal Cre, nCre) injection. (**C**) Immunohistochemical staining of the hippocampal slice with the anti-TREK-1 antibody. (**D**) Quantitative real-time polymerase chain reaction analysis of TREK-1 in the dentate gyrus. The numbers inside each bar indicate the number of sample (**E**) Protein expression of the green fluorescent protein, mCherry, and TREK-1 in the dentate gyrus. Data are presented as means ± standard error of the mean (* *p* < 0.05, ** *p* < 0.01).

**Figure 4 ijms-20-05902-f004:**
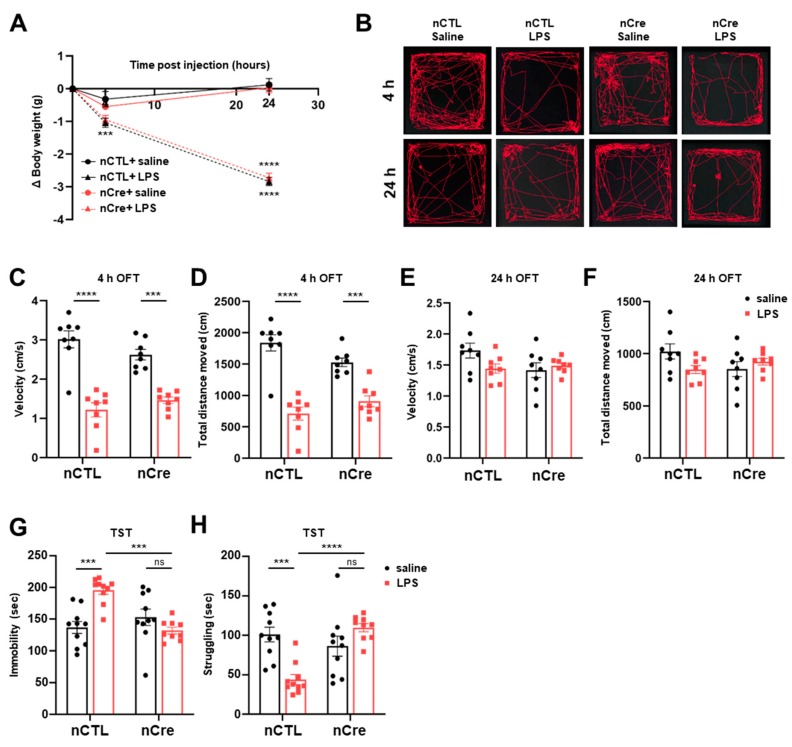
The knockdown of neuronal TREK-1 in the hippocampus exhibited antidepressant behavior. (**A**) Bodyweight changes in the lipopolysaccharide (LPS) or saline-injected groups (neuronal control [nCTL] saline = 8, nCTL LPS = 8, neuronal Cre [nCre] saline = 8, nCre LPS = 8). (**B**) The representative track generated from the open field test on mice injected with LPS or saline. Velocity (**C**) and total distance (**D**) moved time of mice administered with LPS and saline after 4 h (nCTL saline = 8, nCTL LPS = 8, nCre saline = 8, nCre LPS = 8). Vel©ty (**E**) and total distance (**F**) moved time of mice administered with LPS and saline after 24 h (nCTL saline = 8, nCTL LPS = 8, nCre saline = 8, nCre LPS = 8). (**G**) Immobility time and (**H**) struggling time from the tail suspension test on mice injected with LPS after 24 h (nCTL saline = 8, nCTL LPS = 8, nCre saline = 8, nCre LPS = 8). Data are presented as means ± standard error of the mean (** *p* < 0.01, *** *p* < 0.001, n.s, not significant).

**Figure 5 ijms-20-05902-f005:**
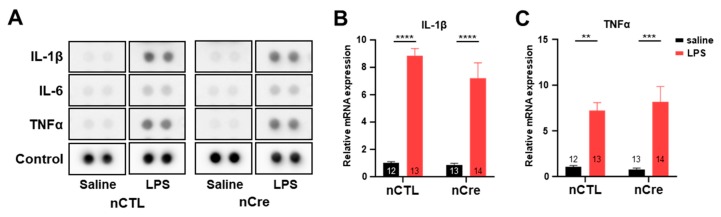
The knockdown of neuronal TREK-1 in the hippocampus did not show cytokine changes under lipopolysaccharide (LPS)-induced depression. (**A**) Multiple cytokine array dot blots in the hippocampal lysates infected with neuronal control or neuronal Cre virus. Quantitative real-time polymerase chain reaction analysis of interleukin 1 beta (**B**) and tumor necrosis factor α (**C**). The numbers inside each bar indicate the number of samples. Data are presented as means ± standard error of the mean (** *p* < 0.01, *** *p* < 0.001, **** *p* < 0.0001).

**Figure 6 ijms-20-05902-f006:**
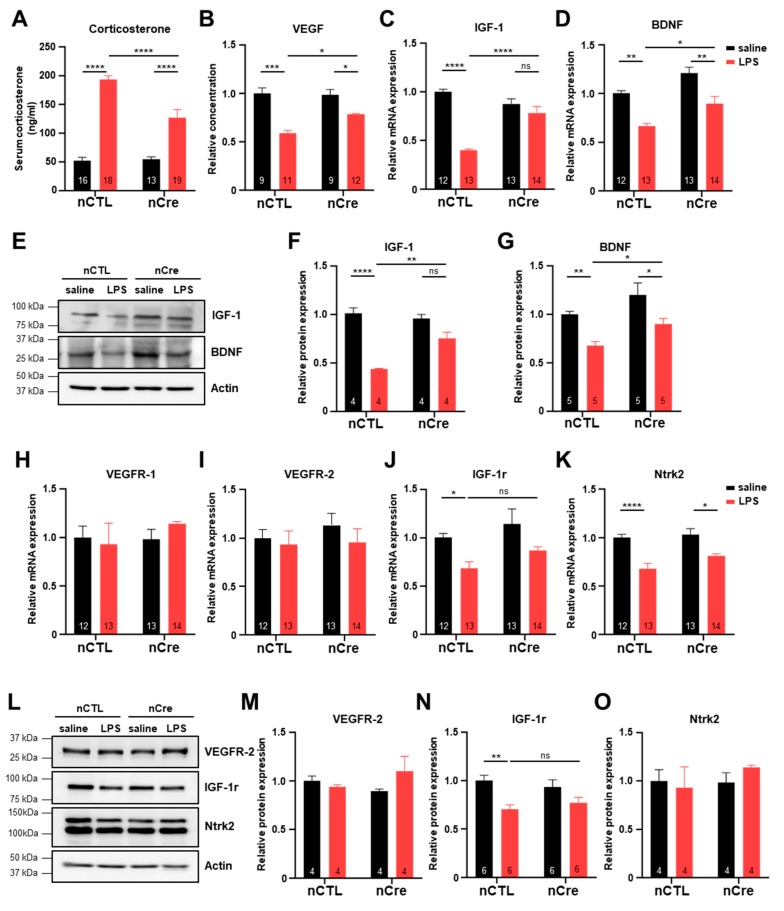
Reduced neurotrophic factor expression in lipopolysaccharide (LPS)-induced depression was preserved in the hippocampal neuronal TREK-1 conditional knockdown mice. (**A**) Serum corticosterone levels in saline or LPS-injected mice. (**B**) Enzyme-linked immunosorbent assay for the vascular endothelial growth factor (VEGF) in mice hippocampal tissues. Quantitative real-time polymerase chain reaction (qRT-PCR) analysis of insulin growth factor-1 (IGF-1) (**C**) and brain-derived neurotrophic factor (BDNF) (**D**). I Western blot analysis from the hippocampal tissues infected with neuronal CTL or neuronal Cre virus (**E**). Corresponding quantified densitometry of the western blotting results, as shown in I for IGF-1 (**F**) and BDNF (**G**). qRT-PCR analysis of VEGFR receptor-1 (**H**), VEGFR receptor-2 (VEGFR-2) (**I**), IGF-1r (**J**), and Ntrk2 (**K**). (**L**) Western blot analysis of hippocampal tissues. Blots were performed using the antibodies indicated at right. Corresponding quantified densitometry of the western blotting results, as shown in (**L**) for VEGFR-2 (**M**), IGF-1r (**N**), and Ntrk2 (**O**). The numbers inside each bar indicate the number of samples. Data are presented as means ± standard error of the mean (* *p* < 0.05, ** *p* < 0.01, *** *p* < 0.001, **** *p* < 0.0001).

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
