# Peer review of "The Knockdown of TREK-1 in Hippocampal Neurons Attenuate Lipopolysaccharide-Induced Depressive-Like Behavior in Mice"

_ijms, 2019, doi:10.3390/ijms20235902_

Round 1
Reviewer 1 Report
In this work a conditional knockdown mouse, with reduced expression of TREK-1 channels in the hippocampus, is used to investigate the role of hippocampal TREK-1 channels in depression. The authors conclude that the reduction of TREK-1 channels in the hippocampus justifies the antidepressant effect that results from inhibiting these channels. They also ensure that the reduction of TREK-1 in the hippocampus is able to prevent the reduction of BDNF, VEGF and IGF1, and therefore the depression that lipopolysaccharides normally produce.
Abstract, L31. The conclusion issued in the last sentence of the abstract is very confusing. What does "neural selective TREK-1" mean? What does "inhibiting decreased expression of" mean? It is not clear which control group is discussed on line 27. The abstract should be completely rewritten and should follow a logical order. In general the text contains many grammatical errors and should be reviewed by a native English speaker. The mechanism by which the reduction of TREK-1 activity in the hippocampus causes a decrease in neurotrophic factors is unclear. The last paragraph of the introduction (L75) contributes significantly to the confusion. An electrophysiological study of hippocampal neurons in the transgenic mouse would increase the interest of this work.
Minor points
Introduction, L80. “…selectively inhibited by the hippocampal”. It should say “in” instead of “by”? Introduction, L82. Please explain the meaning of “…sufficient antidepressant effects.” Results, L90: “Moreover, this system archived cell type-specific gene knockdown mice.” Please explain. Results, L165. This phrase makes no sense: “…to determine whether LPS-induced animals were associated with sickness behaviour since they were associated with sickness and depressive-like behaviours…” Results, L218. The sentence makes no sense. Methods, L326. The sentence “…on deeply anesthetized time of 6-7 weeks old,…” is very confusing.
Author Response
Response to Reviewer 1 Comments
Abstract, L31. The conclusion issued in the last sentence of the abstract is very confusing. What does "neural selective TREK-1" mean? What does "inhibiting decreased expression of" mean? It is not clear which control group is discussed on line 27. The abstract should be completely rewritten and should follow a logical order.
Thank you for your comments. So we rewrote the abstract in a logical order (L18~L32).
In general the text contains many grammatical errors and should be reviewed by a native English speaker.
The first submitted manuscript was reviewed by the English editing service (Editage;EALNB_1) and this revised manuscript was further reviewed by the same service (EALNB_1_3).
The mechanism by which the reduction of TREK-1 activity in the hippocampus causes a decrease in neurotrophic factors is unclear.
TREK-1 is a background potassium channel, and inhibition of TREK-1 in neuronal cells induces depolarization of the resting membrane potential (RMP) and neuronal excitability. Although the mechanism by which TREK-1 inhibition regulates the expression of neurotrophic factors is unknown, neuronal activity increased by inhibition of TREK-1 may play a role in enhancing neurotrophic factor expression by AKT-ERK signaling (L310-314).
The last paragraph of the introduction (L75) contributes significantly to the confusion.
Sorry for the confusion. We rewrote the introduction section (L75) to reduce confusion.
An electrophysiological study of hippocampal neurons in the transgenic mouse would increase the interest of this work.
I agree with your suggestion. If electrophysiology studies are carried out as further studies, we expect to understand much more about TREK-1.
Minor points
Introduction, L80. “…selectively inhibited by the hippocampal”. It should say “in” instead of “by”? Introduction, L82. Please explain the meaning of “…sufficient antidepressant effects.”
Sorry for the confusion. We rewrote the introduction section (L74-81) for better understanding.
Results, L90: “Moreover, this system archived cell type-specific gene knockdown mice.” Please explain.
We rewrote the results section (L87-L89) in detail.
Results, L165. This phrase makes no sense: “…to determine whether LPS-induced animals were associated with sickness behaviour since they were associated with sickness and depressive-like behaviours…”
We rewrote the results section (L167-168) to better explain the purpose of the behavioral experiment.
Results, L218. The sentence makes no sense.
We rewrote the results section (L224-227) to explain the causality of why we looked at corticosterone and neurotrophic factors.
Methods, L326. The sentence “…on deeply anesthetized time of 6-7 weeks old,…” is very confusing.
Sorry for the confusion. We have modified the methods section (L344-346).
Reviewer 2 Report
This manuscript by Kim et al. showed that hippocampal neuronal TREK-1 K+ channel is critical for LPS-induced depression using original TREK-1 conditional knockdown mice and TREK-1 has antidepressant effects by inhibiting decreased expression of neurotrophic factors. The experimental strategies are adequate, however, however, there are some concerns that need to be addressed.
Major concerns:
In summarized results (Fig. 2D,F, Fig. 3D, Fig. 4A & C-F, Fig. 6A-D, F, G-J, Fig. S3C, Fig. S4A-C), the number of samples (n) is completely lacking. All authors should critically review the manuscript, and approve its final version. Inhibition of TREK-1 in neuronal cells induces depolarization-induced Ca2+ influx and activates Ca2+ In addition, TREK-1 is involved in PI3K/AKT signaling and ERK signaling. From the view point of TREK-1 downstream signaling pathways, authors must explain the results of Figure 6 in ‘Discussion’ section. In addition, authors concluded that there are no difference in the expression of neurotrophic factor receptors (Fig. 6G-J). Authors examined the expression levels of their mRNAs alone. The readers will be unconvinced without the results on protein expression levels and/or functional analysis. In Fig. 2D, real-time PCR examination should be performed, similar to the other section.
Major concerns:
5: What is ‘control’? Authors should replace ‘Supple Figure 1-4’ with ‘Supplementary Figure S1-S4’.
Author Response
Response to Reviewer 2 Comments
Major concerns:
In summarized results (Fig. 2D,F, Fig. 3D, Fig. 4A & C-F, Fig. 6A-D, F, G-J, Fig. S3C, Fig. S4A-C), the number of samples (n) is completely lacking. All authors should critically review the manuscript, and approve its final version.
Thanks for helpful comments. We added ‘n’ values for all experimental data
Inhibition of TREK-1 in neuronal cells induces depolarization-induced Ca2+ influx and activates Ca2+ In addition, TREK-1 is involved in PI3K/AKT signaling and ERK signaling. From the view point of TREK-1 downstream signaling pathways, authors must explain the results of Figure 6 in ‘Discussion’ section.
To address this point, we added the following sentence in the discussion section (L310-314).
"Additionally, inhibition of TREK-1 in neuronal cells induces depolarisation-induced Ca2+ influx and activates PI3K/AKT and extracellular signal-regulated kinase (ERK) signalling [46]. Thus, reduction of TREK-1 in the nCre groups of our mice may also increase the basal activity of the AKT and ERK signalling, which may be effective in preventing BDNF reduction in inflammatory responses."
In addition, authors concluded that there are no difference in the expression of neurotrophic factor receptors (Fig. 6G-J). Authors examined the expression levels of their mRNAs alone. The readers will be unconvinced without the results on protein expression levels and/or functional analysis.
We would like to thank the reviewer for pointing this out. Thus, we performed additional experiments to address the reviewer`s comments and added the obtained data in Figure 6 (Western Blot analysis of IGF-1, VEGFR-2r, IGF-1r, and Ntrk2). The results showed that protein expression of neurotrophic receptors (VEGF-2r, IGF-1r, and Ntrk2) are not significantly difference in LPS treatment condition (IGF-1; Figure 6E, F, VEGF-2r; Figure 6L, M, IGF-1r; Figure 6L, N, Ntrk2; Figure 6L, O).
In Fig. 2D, real-time PCR examination should be performed, similar to the other section.
We added the real-time PCR data in Figure 2E.
Major concerns:
5: What is ‘control’?
We apologize for the insufficient explanation about the mean of ‘control’. The ‘Control’ (CTL) groups refer to mice injected only with the fluorescent protein virus (eg. AAV-Ef1a-mCherry or AAV-hSyn-BFP). And ‘Cre’ groups refer to mice injected with fluorescent tagged Cre virus (eg. AAV-EF1a-mCherry-IRES-Cre or AAV-hSyn-BFP-Cre).
Authors should replace ‘Supple Figure 1-4’ with ‘Supplementary Figure S1-S4’.
Thank you for your comment. We have corrected all.
Round 2
Reviewer 1 Report
The authors have satisfactorily answered my questions and have corrected the manuscript properly
Reviewer 2 Report
The manuscript has been adequately improved. The reviewer has no more concerns.